# Functional ability and quality of life in critical illness survivors with intensive care unit acquired weakness: A secondary analysis of a randomised controlled trial

Sabrina Eggmann[1,2]*, Gere Luder[1], Martin L. Verra[1], Irina Irincheeva[3], Caroline H. G. Bastiaenen[2], Stephan M. Jakob[4]

1 Department of Physiotherapy, Insel Group, Inselspital, Bern University Hospital, Bern, Switzerland, 2 Department of Epidemiology, Research Line Functioning and Rehabilitation CAPHRI, Maastricht University, Maastricht, the Netherlands, 3 CTU Bern, University of Bern, Bern, Switzerland, 4 Department of Intensive Care Medicine, Inselspital, Bern University Hospital, University of Bern, Bern, Switzerland

* sabrina.eggmann@insel.ch

## Abstract

### Introduction

Intensive care unit acquired weakness (ICUAW) may contribute to functional disability in ICU survivors, yet performance-based data for general ICU patients are lacking. This study explored functional outcomes of (1) and risk factors for (2) weakness at ICU discharge.

### Methods

Data from a randomised controlled trial that investigated two early exercise regimes in previously independent, ventilated adults (n = 115) without any significant outcome-differences were used for the present analysis. ICUAW was clinically diagnosed in cooperative participants (n = 83) at ICU discharge with the Medical Research Council sum-score (MRC-SS) using a cut-off <48 for moderate or <36 for severe weakness. Primary outcomes were the 6-Minute Walk Test and Functional Independence Measure at hospital discharge. Secondary outcomes included health-related quality of life after six months. Risk factors during the ICU stay were explored for their effect on MRC-SS with linear regression.

### Results

Functional outcomes and length of hospital stay significantly differed in patients with severe, moderate to no weakness (6-Minute Walk test: p = 0.013; 110m [IQR 75–240], 196m [90–324.25], 222.5m [129–378.75], Functional Independence Measure: p = 0.001; 91[IQR 68–101], 113[102.5–118.5], 112[97–123], length of stay after ICU discharge: p = 0.008; 20.9d [IQR 15.83–30.73], 16.86d [13.07–27.10], 11.16d [7.35–19.74]). However, after six months participants had similar values for quality of life regardless of their strength at ICU discharge (Short-Form 36 sum-scores physical health: p = 0.874, mental health: p = 0.908). In-bed

**Data Availability Statement:** The dataset and the analysis script are available from the "Bern Open Repository and Information System" database

(https://boris.unibe.ch/140381/). The source code to reproduce the publication's results is available at https://github.com/CTU-Bern/FuncAbil_icuaw.

**Funding:** This research has been funded with the PhD Grant 2018 from the Swiss Foundation for Physiotherapy Science received by SE. The foundation had no role in study design, data collection and analysis, decision to publish, or preparation of the manuscript.

**Competing interests:** SE, GL, MLV, CHGB declare that they have no competing interests. II is affiliated with CTU Bern, University of Bern, which has a staff policy of not accepting honoraria or consultancy fees. However, CTU Bern is involved in design, conduct, or analysis of clinical studies funded by not-for-profit and for-profit organizations. In particular, pharmaceutical and medical device companies provide direct funding to some of these studies. For an up-to-date list of CTU Bern's conflicts of interest see http://www.ctu.unibe.ch/research/declaration_of_interest/index_eng.html. SMJ reports the following potential conflicts of interest: The Department of Intensive Care Medicine has, or has had in the past, research contracts with Orion Corporation, Abbott Nutrition International, B. Braun Medical AG, CSEM SA, Edwards Lifesciences Services GmbH, Kenta Biotech Ltd, Maquet Critical Care AB, Omnicare Clinical Research AG and research & development/consulting contracts with Edwards Lifesciences SA, Maquet Critical Care AB, and Nestlé. The money was paid into a departmental fund; SMJ received no personal financial gain. The Department of Intensive Care Medicine has received unrestricted educational grants from the following organizations for organizing a quarterly postgraduate educational symposium, the Berner Forum for Intensive Care (until 2015): Fresenius Kabi, gsk, MSD, Lilly, Baxter, astellas, AstraZeneca, B | Braun, CSL Behring, Maquet, Novartis, Covidien, Nycomed, Pierre Fabre Pharma AG (formerly known as RobaPharm), Pfizer, Orion Pharma, Bard Medica S.A., Abbott AG, Anandic Medical Systems. The Department of Intensive Care Medicine has received unrestricted educational grants from the following organizations for organizing bi-annual postgraduate courses in the fields of critical care ultrasound, management of ECMO and mechanical ventilation: Pierre Fabre Pharma AG (formerly known as RobaPharm), Pfizer AG, Bard Medica S.A., Abbott AG, Anandic Medical Systems, PanGas AG Healthcare, Orion Pharma, Bracco, Edwards Lifesciences AG, Hamilton Medical AG, Fresenius Kabi (Schweiz) AG, Getinge Group Maquet AG, Dräger Schweiz AG, Teleflex Medical GmbH. No author was

immobilisation was the most significant factor associated with weakness at ICU discharge in the regression models (MRC-SS: -24.57(95%CI [-37.03 to -12.11]); p<0.001).

## Conclusions

In this general, critically ill cohort, weakness at ICU discharge was associated with short-term functional disability and prolonged hospital length of stay, but not with quality of life, which was equivalent to the values for patients without ICUAW within six months. Immobilisation may be a modifiable risk factor to prevent ICUAW. Prospective trials are needed to validate these results.

## Trial registration

German Clinical Trials Register (DRKS) identification number: DRKS00004347, registered on September 10, 2012.

## Introduction

Recent years have seen increased survival rates for severely ill patients admitted to intensive care units (ICU) [1, 2]. However, long-term functional disability as well as cognitive and mental health impairment are common in ARDS or sepsis survivors [3–6], leading to poor quality of life and a substantial five-year mortality [7].

Muscle weakness may be a key contributor to the persisting disability of these survivors [8, 9]. However, there is a paucity of data to support a negative impact of ICU-acquired weakness (ICUAW) on physical functioning [10]. Moreover, ICUAW may also be present in less severely ill patients [11]. The few studies that investigated functional outcomes in ICUAW survivors were limited to specific subgroups (e.g. ARDS) [12, 13], lacked performance-based measurements [14–16] or were conducted after the post-acute phase thereby including only the weakest patients [17]. More research on the physical consequences of ICUAW is therefore highly needed to advance early treatment and to prevent long-term disability after critical illness.

Additionally, early identification of persons at risk for ICUAW is necessary for targeted therapeutic or preventive interventions. Postulated risk factors for ICUAW are multiorgan failure [18], increased systemic inflammation [19], female sex [20], duration of mechanical ventilation [21] or bed rest [12]. However, the strength of these risk factors' association with ICUAW remains uncertain and requires further investigation [10].

This exploratory study therefore aimed first to investigate functional outcomes at hospital discharge and health-related quality of life after six months in critically ill patients with severe, moderate or no ICUAW at ICU discharge, and second to explore the role of early risk factors for reduced muscle strength at ICU discharge in mechanically ventilated, critically ill adults.

## Methods

### Design and setting

Data for this secondary analysis were collected as part of a randomised controlled trial that compared very early endurance training and mobilisation to usual care in mechanically ventilated, critically ill adults [22]. The trial was prospectively registered (DRKS00004347) and conducted in the mixed ICU of a large, tertiary academic centre in Switzerland (Department of

employed by any of these companies. None of the companies played a role in this study. Our relationship with the companies does not alter our adherence to PLOS ONE policies on sharing data and materials.

Intensive Care Medicine, Inselspital, Bern University Hospital) between October 8, 2012 and April 5, 2016. No significant differences were found for the primary or secondary outcomes, whereby participants of the experimental group appeared to have better mental health at six months after hospital discharge. A high ICUAW incidence in the whole cohort led to the research questions of this secondary analysis, which was subsequently approved by the local ethics committee (Ethics Committee of Canton Bern) in February 2019 (ID 2019–00156). During the randomised controlled trial, written informed consent was obtained from the next of kin within 72 hours after randomisation and a written informed consent from each patient as early as possible. The need for further consent was waived by the ethics committee.

## Study population and management

Eligible participants were older than 18 years, had been independent before the episode of critical illness and were expected to remain on mechanical ventilation for at least 72 hours. Participants with pre-existing muscle weakness, preceding hospital stay of more than 10 days duration, contraindications to cycling, enrolment in another trial, palliative care, diagnoses that precluded walking at hospital discharge, or insufficient command of German or French were excluded.

All patients were managed by targeted, light sedation and protocol-guided weaning [23]. Standard ICU care also included a nutrition protocol and regular assessments of energy expenditure using indirect calorimetry [24]. Physiotherapy started within 48 hours of ICU admission with two different exercise regimes that included usual early mobilisation versus early, progressive endurance and resistance training in addition to usual early mobilisation. All participants received standard therapy after ICU discharge such as cycling, walking, strengthening, breathing and functional exercises. Further details on study interventions and procedures have been published elsewhere [25].

## Data collection and measurements

Baseline demographics and clinical characteristics were collected at study enrolment by the study nurse responsible. Functional measurements were administered by trained physiotherapists blinded to initial group allocation. Muscle strength was assessed with the Medical Research Council sum-score (MRC-SS) at ICU discharge. The MRC-SS evaluates strength in three bilateral muscles of the upper and lower extremities from 0 (no contraction) to 5 (normal strength) with a maximal summed score of 60 [26]. A cut-off of less than 48 points was used to clinically diagnose ICUAW [27]. To further differentiate between moderate and severe weakness, participants who scored less than 36 were considered severely weak [28]. Valid and reliable strength assessments are dependent upon sufficient cooperation by the participant [29]. Accordingly, the MRC-SS was only performed in participants who were able to follow at least 3 out of 5 standardised commands [20]. In the case of an ICU re-admission within the same hospital stay, the last available MRC-SS was used for analysis to account for subsequent weakness. ICU re-admissions were subject to the same study procedures and data collection as described previously.

The primary functional outcomes for this analysis were chosen as per the original trial: functional independence evaluated with the Functional Independence Measure (FIM) [30] and functional capacity assessed with the 6-Minute Walk Test (6MWT) [31] at hospital discharge. Additional secondary outcomes of interest were FIM at ICU discharge, Timed 'Up & Go' [32] at hospital discharge, hospital length of stay and discharge destination, tracheostomy incidence, ICU readmissions, hospital and 6-month mortality as well as participants' health-related quality of life determined with the Short Form 36 (SF-36) [33] six months after hospital

discharge. We further explored whether functional performance at hospital discharge might be useful to predict 6-month quality of life. See supporting information for a detailed timetable (S1 Table).

## Statistical analysis

The two randomised groups did not differ with regard to the primary or secondary outcomes, consequently, we considered the cohort as one single cohort. Patient demographics and characteristics were summarised with descriptive statistics. Continuous data could not be assumed normally distributed and are therefore presented as medians with interquartile ranges (IQR: first (25%) quartile to third (75%) quartile). Categorical data are given as numbers with percentages.

We hypothesised that participants with no ICUAW would achieve better functional performance at hospital discharge and a higher quality of life after six months when compared to participants with moderate to severe ICUAW. The null hypothesis of equal distributions in functional outcomes for patients diagnosed with no (MRC-SS >48), moderate (MRC-SS 36–47) or severe ICUAW (MRC-SS <36) was tested against the alternative hypothesis of an ordered relationship with the Cuzick test [34]. For example, when applying the Cuzick test to '6MWT', the null hypothesis of equal distributions was tested against the following alternative hypothesis: severe weakness < no weakness and severe weakness ≤ moderate weakness ≤ no weakness, and when applying the Cuzick test to 'hospital length of stay', the alternative hypothesis was severe weakness > no weakness and severe weakness ≥ moderate weakness ≥ no weakness. In baseline variables the ordered relationship among groups did not make sense, therefore, the baseline measurements for the three MRC-SS groups were compared with the non-parametric Kruskal-Wallis test. Categorical data were analysed with Pearson's Chi-Squared test. Baseline characteristics of participants with complete versus missing MRC-SS values were investigated for baseline differences because of the prerequisite of cooperation for a valid MRC-SS assessment. Correlations were investigated with non-parametric Spearman correlation coefficients. Considering the aims of this analysis, multiple imputations were not performed as expected extrapolation would have been applied to patients who died or were unable to follow commands and thus unable to be functionally active. However, to account for the limited sample size, we conducted a sensitivity analysis to compare the data of participants without ICUAW to the pooled data of participants with severe and moderate ICUAW with the non-parametric Mann-Whitney-U tests for between-group comparisons.

The ICU risk factors for muscle weakness at ICU discharge to be explored were identified from previous evidence [35] and chosen based on the available data by favouring uncorrelated risk factors. We assumed that our analysis would reveal some influence of the chosen risk factors for reduced muscle strength at ICU discharge regardless of initial randomisation. To avoid overfitting, we restricted the factors to six (with a rule of thumb of a minimum 10 observations per factor [36]): Sequential Organ Failure Assessment (SOFA) at study inclusion [18, 19], gender [20], length of ICU stay [37], previous limitation in activities of daily living (ADL) [38], mobility level in the ICU (in-bed, edge-of-bed, out-of-bed) [12], and initial randomisation group (control or experimental) in order to exclude a confounding effect. We investigated the effect of all risk factors on MRC-SS as crude (with simple linear regression) and adjusted for the totality of considered factors (with multivariable linear regression). Two sensitivity analyses were conducted with all listed risk variables excluding either randomisation group (potential confounder) or previous ADL limitation (due to missing data).

Data analysis was performed with R version 3.5.3 (2019-03-11) and SPSS (IBM SPSS Statistics Version Premium GradPack 24). The level of significance was set at p<0.05 (two-tailed).

Due to the exploratory nature of this analysis, we decided a priori that the significance threshold would not be adjusted for multiple testing.

## Results

Of the 115 participants who were enrolled in the randomised controlled trial, 83 (72%) completed an MRC-SS at ICU discharge (Fig 1). Main reasons for missing MRC-SS values were ICU-death (14%) and patients' inability to follow commands (11%), while 5 patients (4%) were missed for follow-up at ICU discharge. Among the assessed patients, ICUAW was common with 17 (20%) severe, 32 (39%) moderate, and 34 (41%) no weakness, respectively. The patients in the three MRC-SS groups did not differ significantly in their baseline demographics and characteristics, except for a higher prevalence of liver disease in the severely weak group (Table 1). In contrast, participants with missing MRC-SS values differed in respect to APACHE II scores (completed 21 [IQR 17–26]; missing 27 [21.5–30]), SOFA (completed 8 [6–10]; missing 10 [8 to14.25]), and frequency of liver disease (completed 7 (9%); missing 8 (26%)) (S2 Table and S3 Table).

### Functional outcomes, hospital variables and quality of life

The differences between severe, moderate and no weakness for the primary and secondary outcomes are presented in Table 2. Weakness at ICU discharge was significantly associated with functional disability at hospital discharge and subsequent length of hospital stay, but not with health-related quality of life after six months. The distribution between the three MRC-SS groups for the two primary outcomes is illustrated in Fig 2, rejecting the null-hypothesis of equal distributions. Thus, for both 6MWT and FIM participants with 'severe weakness' performed less well compared to 'no weakness', performance in 'severe weakness' was similar or worse to 'moderate weakness', which was also similar to or worse than 'no weakness'. The achieved percentages of age-predicted values for the 6MWT were also significantly different (p = 0.006: Cuzick test) with 36% (IQR 19–65), 55% (IQR 25–94) and 74% (IQR 39–89) of

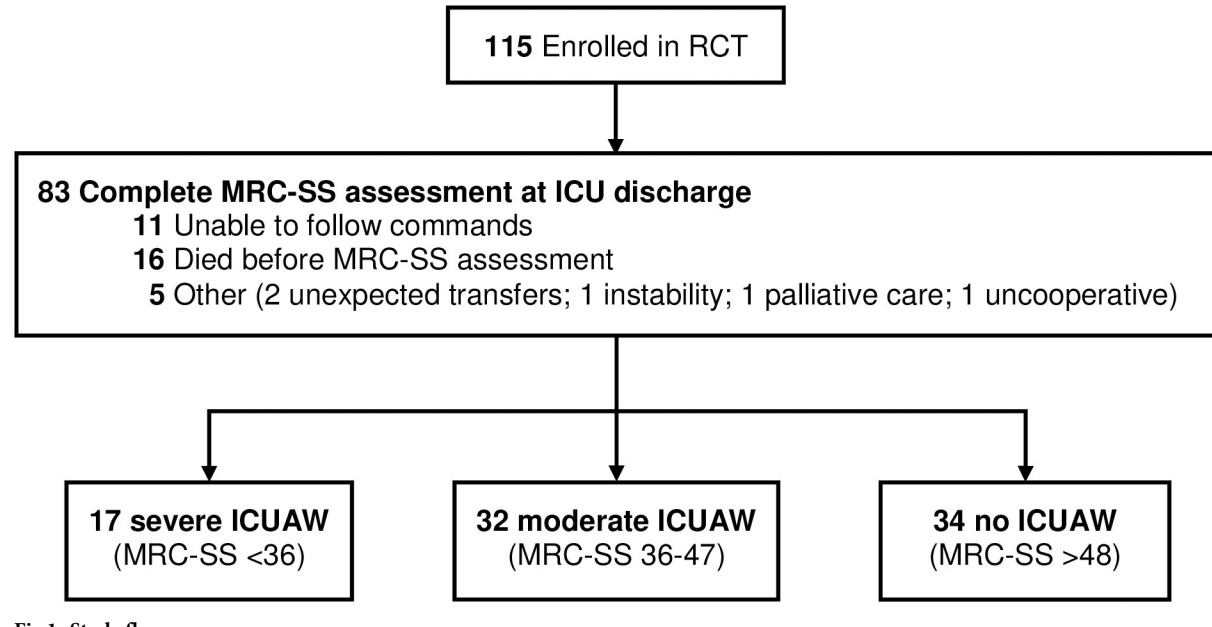

**Fig 1. Study flow.**

**Table 1. Baseline demographics for participants with complete MRC-SS assessment.**

| Variables | n | All with complete MRC-SS | n | Severe weakness (MRC-SS <36) | n | Moderate weakness (MRC-SS 36–48) | n | No weakness (MRC-SS >48) | p-value |
|---|---|---|---|---|---|---|---|---|---|
| ICUAW incidence (MRC-SS<48) | 83 | 49 (59%) | 17 | 17 (100%) | 32 | 32 (100%) | 34 | 0 (0%) | |
| MRC-SS at ICU discharge (0–60) | 83 | 45 [38.5–54] | 17 | 26 [20–31] | 32 | 42 [40–45] | 34 | 55.5 [50–58] | |
| Randomized to non-standard intervention | 83 | 40 (48%) | 17 | 11 (65%) | 32 | 12 (38%) | 34 | 17 (50%) | 0.186 |
| Age (years) | 83 | 67.5 [55.55–75.4] | 17 | 68.1 [65.3–74.9] | 32 | 67.85 [55.9–74.75] | 34 | 60.75 [45.08–76.3] | 0.284 |
| Gender (male) | 83 | 52 (63%) | 17 | 9 (53%) | 32 | 17 (53%) | 34 | 26 (76%) | 0.095 |
| BMI (kg/m$^2$) | 83 | 26.2 [23.6–31.35] | 17 | 27.8 [22.2–34] | 32 | 27 [23.98–29.85] | 34 | 25.65 [23.9–31.5] | 0.984 |
| Weight (kg) | 83 | 80 [66–90] | 17 | 85 [62–95] | 32 | 79.2 [66.5–85] | 34 | 80 [68.25–90] | 0.783 |
| APACHE II score (0–71) [a] | 83 | 21 [17–26] | 17 | 23 [18–26] | 32 | 21.5 [18–26.25] | 34 | 20 [16.25–23.75] | 0.483 |
| SOFA score (0–24) [b] | 83 | 8 [6–10] | 17 | 9 [7–10] | 32 | 8 [7–10.25] | 34 | 6.5 [5–10] | 0.065 |
| ICU days until study inclusion | 83 | 1.71 [0.85–2.57] | 17 | 1.76 [1.24–2.63] | 32 | 1.84 [1.05–2.61] | 34 | 1.51 [0.8–2.08] | 0.518 |
| ICU length of stay at original hospital (days) | 83 | 5.93 [4.43–10.26] | 17 | 6.23 [4.73–14.35] | 32 | 6.56 [4.28–11.88] | 34 | 5.63 [3.50–7.89] | 0.210 |
| **ICU diagnosis on ICU admission** | | | | | | | | | |
| Gastroenterology | 83 | 11 (13%) | 17 | 2 (12%) | 32 | 6 (19%) | 34 | 3 (9%) | 0.680 |
| Heart surgery | | 18 (22%) | | 5 (29%) | | 6 (19%) | | 7 (21%) | |
| Hemodynamic insufficiency | | 16 (19%) | | 3 (18%) | | 6 (19%) | | 7 (21%) | |
| Neurology / neurosurgery | | 4 (5%) | | 1 (6%) | | 1 (3%) | | 2 (6%) | |
| Other | | 1 (1%) | | 0 (0%) | | 0 (0%) | | 1 (3%) | |
| Other surgery | | 11 (13%) | | 3 (18%) | | 4 (12%) | | 4 (12%) | |
| Respiratory insufficiency | | 20 (24%) | | 2 (12%) | | 9 (28%) | | 9 (26%) | |
| Trauma | | 2 (2%) | | 1 (6%) | | 0 (0%) | | 1 (3%) | |
| **Comorbidities on ICU admission** | | | | | | | | | |
| Restricted in activities of daily living (ADL) | 80 | 8 (10%) | 16 | 3 (19%) | 32 | 4 (12%) | 32 | 1 (3%) | 0.196 |
| NYHA symptoms (stage 2 to 4) | 80 | 36 (45%) | 16 | 6 (38%) | 32 | 18 (56%) | 32 | 12 (38%) | 0.256 |
| Dyspnoea symptoms | 80 | 20 (25%) | 16 | 2 (12%) | 32 | 7 (22%) | 32 | 11 (34%) | 0.223 |
| Hematologic malignancy | 80 | 3 (4%) | 16 | 0 (0%) | 32 | 0 (0%) | 32 | 3 (9%) | 0.097 |
| Immuno-suppression | 80 | 11 (14%) | 16 | 3 (19%) | 32 | 3 (9%) | 32 | 5 (16%) | 0.622 |
| Liver disease | 80 | 7 (9%) | 16 | 4 (25%) | 32 | 2 (6%) | 32 | 1 (3%) | 0.033 |
| Chronic dialysis | 80 | 0 (0%) | 16 | 0 (0%) | 32 | 0 (0%) | 32 | 0 (0%) | |

[a] at ICU admission

[b] at study inclusion

Data are presented as median [IQR 25% - 75%] or frequencies (%). Analysis for continuous variables was performed with the Kruskal–Wallis test for the null hypothesis of equal distributions in the three groups, and for categorical and binary variables with Pearson's Chi-Squared test with the null hypothesis of independence between the tested condition and MRC-SS groups.

**Abbreviations:** NYHA = New York Heart Association, BMI = Body Mass Index, APACHE = Acute Physiology and Chronic Health Evaluation, SOFA = Sequential Organ Failure Assessment

predicted normative values [39] for severe, moderate and no weakness, respectively. Fig 3 further illustrates the distribution between the three MRC-SS groups for the Timed 'Up & Go' test and hospital length of stay after ICU discharge, while Fig 4 illustrates the distribution for the SF-36 physical and mental health sum-scores. Sensitivity analysis comparing all participants with ICUAW versus non-ICUAW similarly confirmed a significant association of ICUAW with functional disability and prolonged hospital stay, and lack of association with 6-month health-related quality of life (S4 Table). Functional performance at hospital discharge

**Table 2. Primary and secondary outcome-comparisons per MRC-SS group.**

| Variable | n | All with complete MRC-SS | n | Severe weakness | n | Moderate weakness | n | No weakness | p-value |
|---|---|---|---|---|---|---|---|---|---|
| **Primary outcomes at hospital discharge** | | | | | | | | | |
| 6MWT (m) | 73 | 185 [95–320] | 17 | 110 [75–240] | 28 | 196 [90–324.25] | 28 | 222.5 [129–378.75] | 0.013 |
| FIM (18–126) | 73 | 110 [92–119] | 17 | 91 [68–101] | 27 | 113 [102.5–118.5] | 29 | 112 [97–123] | 0.001 |
| **Secondary ICU and hospital outcomes** | | | | | | | | | |
| FIM at ICU discharge (18–126) | 83 | 36 [26.5–47.5] | 17 | 24 [21–34] | 32 | 31 [26.5–46] | 34 | 41.5 [35–57.5] | <0.001 |
| Timed 'Up & Go 'test (s) at hospital discharge | 57 | 19 [11.4–25] | 14 | 23.25 [20.25–34] | 21 | 18.7 [12.6–27] | 22 | 14 [8–23.25] | 0.013 |
| Hospital length of stay after ICU discharge (days) | 83 | 16.87 [11.16–26.92] | 17 | 20.9 [15.83–30.73] | 32 | 16.86 [13.07–27.10] | 34 | 11.16 [7.35–19.74] | 0.008 |
| **SF-36: quality of life after 6 months** | | | | | | | | | |
| Physical functioning (0–100) | 54 | 75 [46.25–85] | 14 | 72.5 [55–80] | 15 | 70 [37.5–85] | 25 | 75 [45–90] | 0.449 |
| Role physical (0–100) | 52 | 25 [0–50] | 13 | 50 [25–75] | 14 | 25 [0–43.75] | 25 | 25 [0–50] | 0.583 |
| Bodily pain (0–100) | 54 | 74 [51.25–100] | 14 | 77 [53.75–100] | 15 | 70 [41–92] | 25 | 80 [62–100] | 0.595 |
| General health (0–100) | 52 | 61 [45.75–73.25] | 14 | 58.5 [47–70.75] | 13 | 50 [40–57] | 25 | 67 [52–77] | 0.164 |
| Vitality (0–100) | 53 | 55 [40–70] | 14 | 60 [51.25–73.75] | 14 | 50 [30–55] | 25 | 55 [50–70] | 0.640 |
| Social functioning (0–100) | 53 | 75 [50–100] | 14 | 87.5 [53.12–100] | 14 | 75 [53.12–96.88] | 25 | 75 [62.5–100] | 0.982 |
| Role emotional (0–100) | 52 | 66.67 [33.33–100] | 12 | 100 [33.33–100] | 15 | 33.33 [0–83.34] | 25 | 100 [33.33–100] | 0.795 |
| Mental health (0–100) | 52 | 76 [68–85] | 14 | 82 [69–87] | 14 | 70 [61–79] | 24 | 82 [71–88] | 0.659 |
| Physical health (sum-score) | 49 | 42.6 [34.76–48.23] | 12 | 43.19 [33.11–48.5] | 13 | 42.92 [27.67–47.39] | 24 | 42.18 [36.49–48.75] | 0.874 |
| Mental health (sum-score) | 49 | 50.09 [44.4–56.19] | 12 | 51.3 [44.56–58.08] | 13 | 48 [37.67–51.08] | 24 | 51.86 [46.18–56.19] | 0.908 |

Data are presented as median [IQR 25% - 75%] or frequencies (%). Only effectively measured data were analysed. Categorical and binary variables testing was performed with Pearson's Chi-Squared test (the null hypothesis is independence between the tested condition and the MRC-SS groups). Continuous variables testing was performed with the non-parametric Cuzick test (the null hypothesis is equal distributions in the three groups against the alternative non-inferiority or non-superiority). SF-36 (version 2): worst score: 0, best score: 100, sum-score: T-values where the population mean is 50 and the SD is 10; based on US-population 1990. German norm-based (1994) standardized sum-scores (T-values) for SF-36 were similar to the US-population (data shown in sensitivity analysis in S4 Table).

**Abbreviations:** 6MWT = 6-Minute Walk Test, FIM = Functional Independence Measure, SF-36 = Short Form 36 questionnaire

was not associated with quality of life (S5 Table). Hospital discharge destinations did not significantly differ between the three MRC-SS groups (p = 0.321): Severely weak patients (n = 17) were predominantly discharged to a rehabilitation facility (65%), moderately weak patients (n = 32) to rehabilitation (47%) or an external hospital (25%) and non-weak participants (n = 34) to rehabilitation (53%) or home (29%). Similarly, there was no difference in the MRC-SS groups for ICU readmissions (p = 0.264), tracheostomy incidence (p = 0.630) or hospital and 6-month mortality (p = 0.362).

## Risk factors

Table 3 presents the ICU risk factors that were significantly associated with ICUAW at ICU discharge. In-bed immobilisation and female gender remained significantly associated with low MRC-SS scores in sensitivity analyses (S6 Table). When the ADL variable was removed from the model due to the reduced sample size, length of ICU stay became marginally associated with MRC-SS (-0.28 95%-CI [-0.55 to -0.01], p = 0.049).

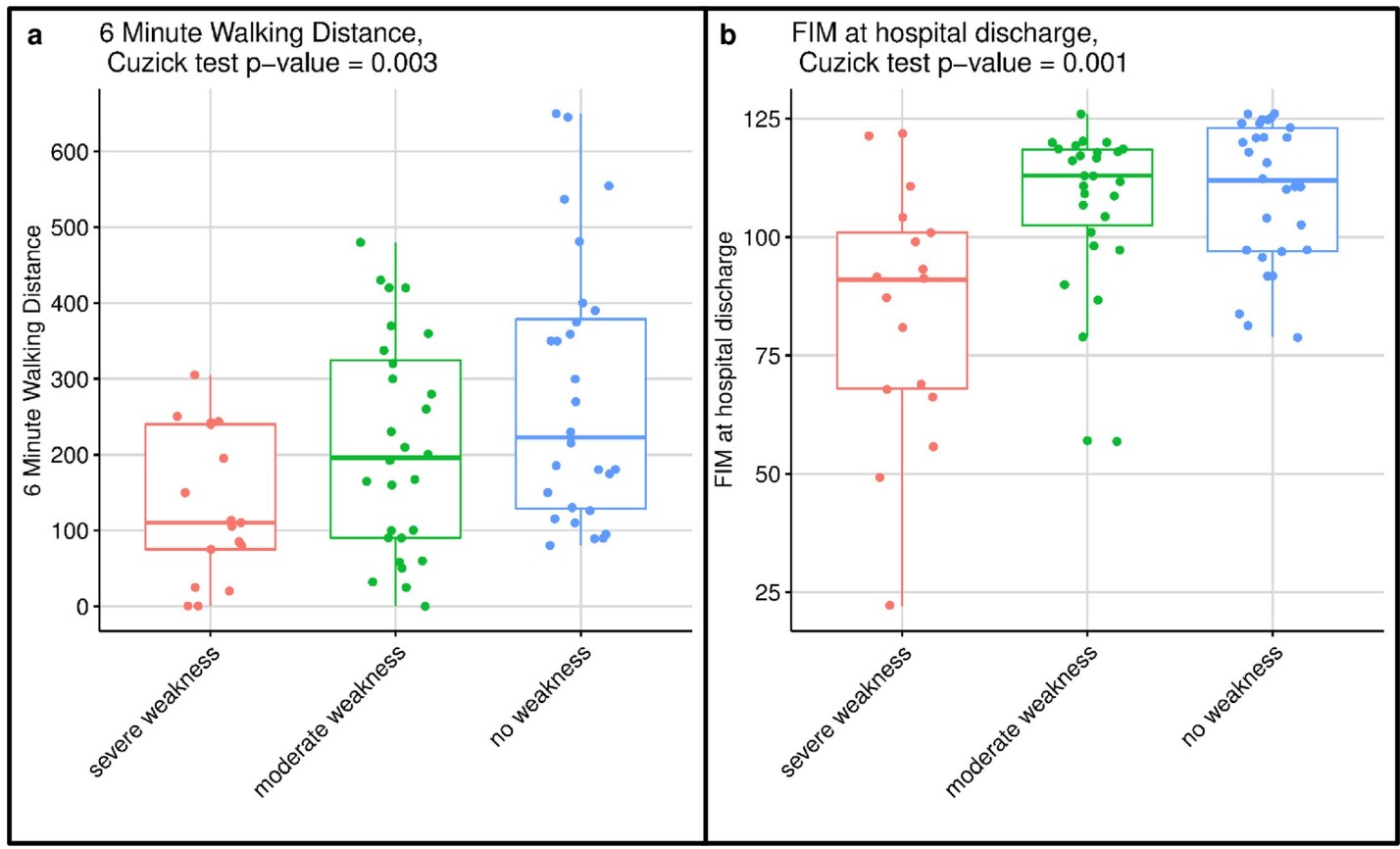

**Fig 2. Illustration of the two primary outcomes per MRC-SS group.** Illustration of the 6MWT (a) and the FIM (b) at hospital discharge with non-parametric Cuzick test clearly rejecting the null hypothesis in favour of alternative: values for severe weakness < no weakness and for severe weakness ≤ moderate weakness vales ≤ no weakness.

## Discussion

In this secondary analysis, participants without ICUAW at ICU discharge had better functional outcomes at hospital discharge with shorter length of hospital stays when compared to participants with moderate and, especially, severe ICUAW. However, contrary to our hypothesis, quality of life after six months was similar in participants with severe, moderate or no ICUAW and not associated with functional performance in the hospital. While mental health sum-scores reached normative values, physical health sum-scores continued to be reduced in all three MRC-SS groups. The risk factor most associated with weakness at ICU discharge was in-bed immobilisation. Female gender or reduced mobility levels were further risk factors of ICUAW. These results provide useful information about the functional ability and subsequent health-related quality of life in a general, mechanically ventilated ICU population after a critical illness period, but need to be validated in prospective studies.

There are few available studies with which to compare our findings on the functional outcomes of ICUAW patients at hospital discharge. Hermans et al. [14] reported no difference in the 6MWT in matched weak and non-weak patients. However, high dropout rates limited this conclusion and results were not confirmed in sensitivity analyses. Although not performed at hospital discharge, Fan et al. [12] investigated the 6MWT over 24 months in ARDS patients. After three months, they found no difference between ICUAW versus non-ICUAW patients, yet for all later time-points, the distance walked was significantly shorter in participants with

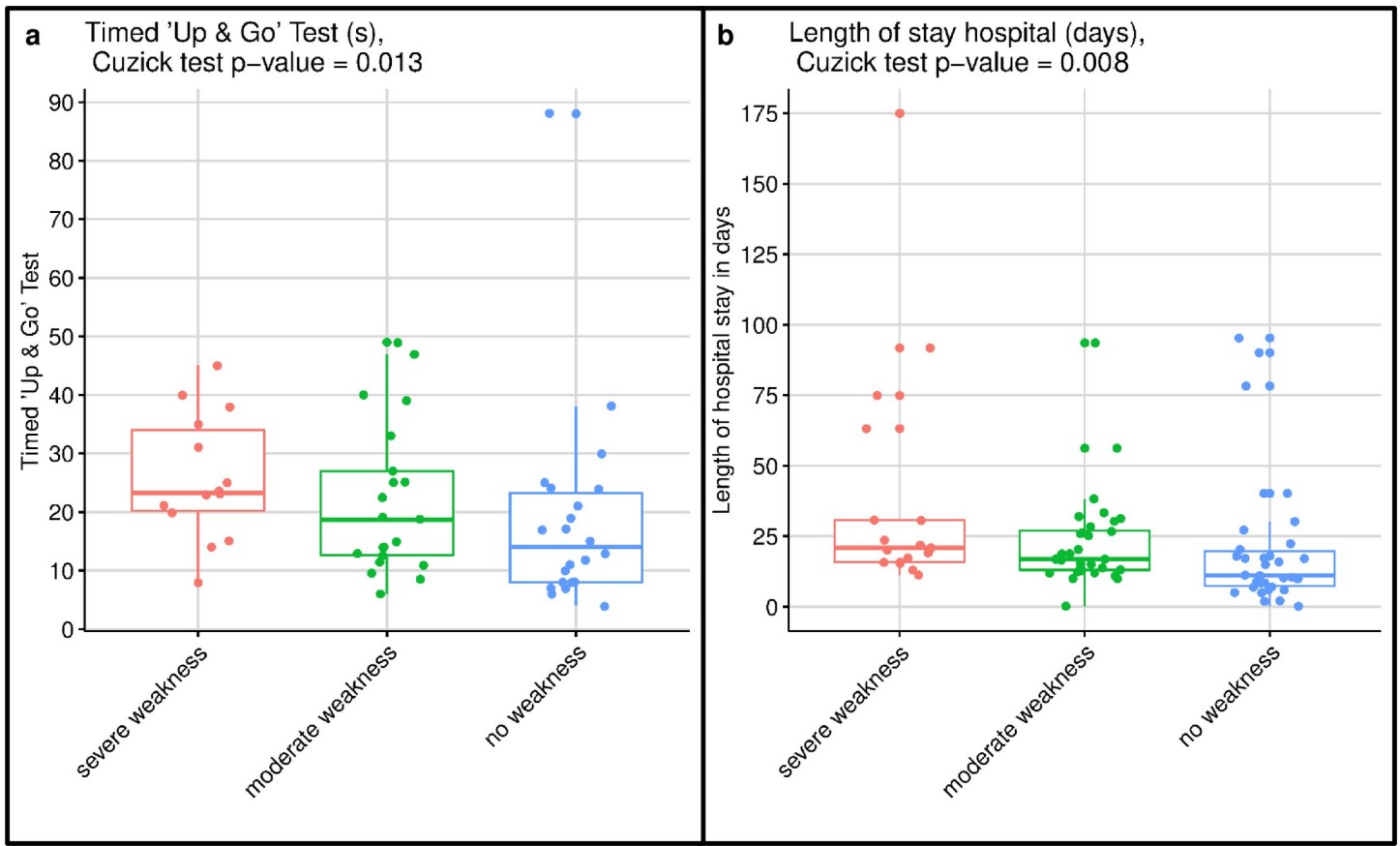

**Fig 3. Illustration of the Timed 'Up & Go' test and hospital length of stay per MRC-SS group.** Illustration of the Timed 'Up & Go' test (a) and hospital length of stay after ICU discharge (b) with non-parametric Cuzick test clearly rejecting the null hypothesis in favour of alternative: values for severe weakness > no weakness and severe weakness ≥ moderate weakness ≥ no weakness.

ICUAW. Sidiras et al. [40] also described significant differences of the FIM at hospital discharge in ICUAW versus non-ICUAW patients. Yet, while values for non-ICUAW patients were comparable to our results, ICUAW patients were substantially less independent in their study (65 [IQR 53–87] in [40] versus 105.5 [88.5–117.5]). This might explain their results for 3- and 6-month follow-up, which were also significantly lower in participants with ICUAW, whereas our patients showed similar health-related quality of life. Overall our findings reinforce the rather weak evidence that ICUAW is associated with functional disability which may lead to less independence at hospital discharge. In turn, loss of independence might be the cause of prolonged length of hospital stays and increased health-care costs for patients with ICUAW [14], emphasising the need to make recovery a priority in acute care [41].

When comparing our results for health-related quality of life with the existing literature, we found some discrepancies. For example, weak and non-weak ARDS survivors differed significantly in their SF-36 physical function sum-scores at six months in a US cohort study [12]. Similarly, a Dutch study [15] reported significant differences in the SF-36 domain 'physical functioning' between ICUAW (45 [IQR 30–70]) and non-ICUAW (75 [50–90]) in participants from a general ICU population. In our study, regardless of initial weakness, participants presented with similar median values in the 'physical functioning' domain (ICUAW 70 [IQR 50–80] versus non-ICUAW 75 [45–90], p = 0.375) as the participants without ICUAW in the Dutch study [15]. Finally, in a Greek trial, participants with ICUAW scored significantly lower

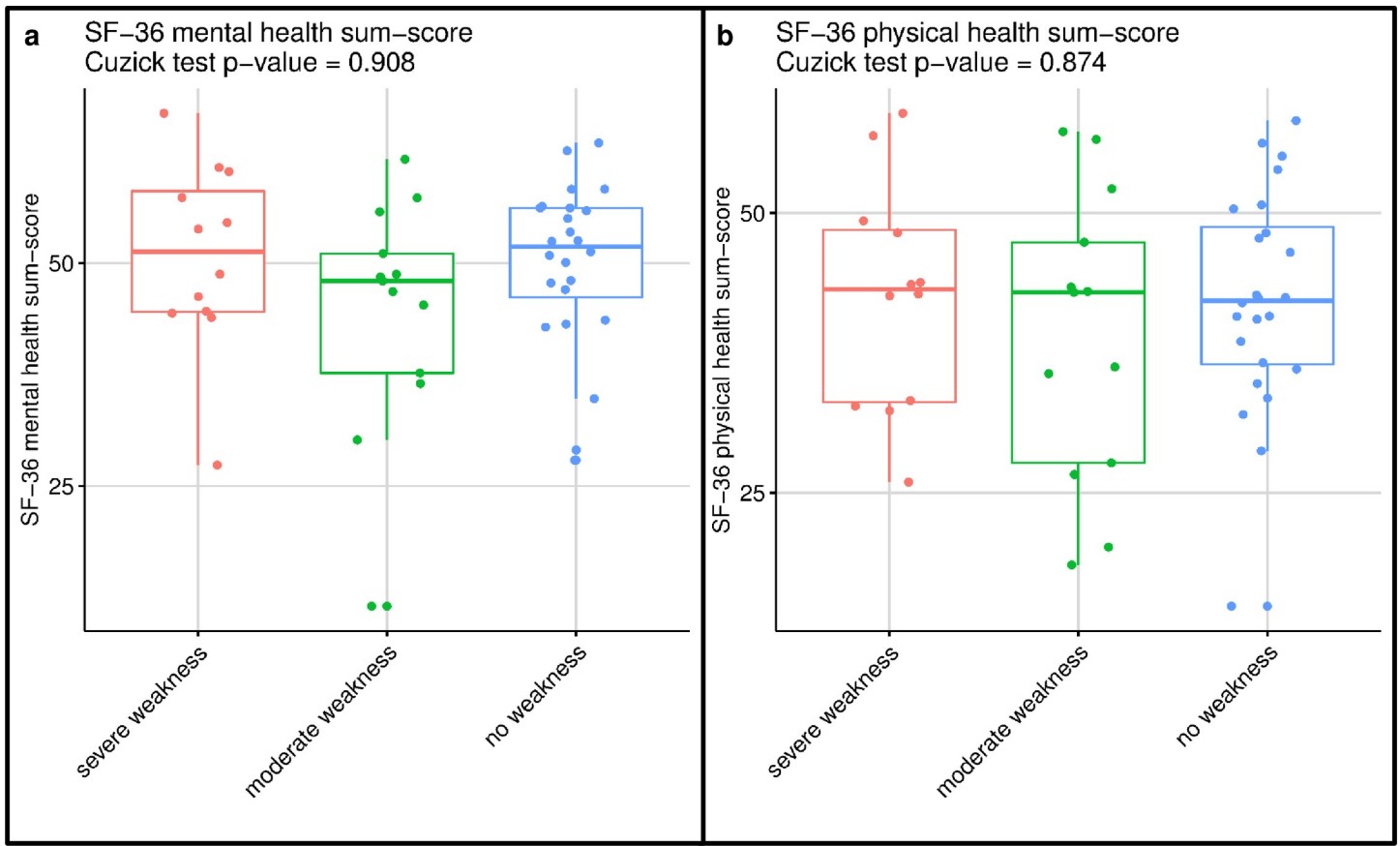

**Fig 4. Illustration of the SF-36 physical and mental health sum-scores per MRC-SS group.** Illustration of the SF-36 physical health sum-score (a) and mental health sum-score (b) with non-parametric Cuzick test accepting the null hypothesis of equal distributions.

in the SF-36 domains of 'general health', 'pain', 'physical functioning' and 'role physical' than non-ICUAW [40]. The apparent recovery of the very weak patients in our study was therefore surprising and may imply a different pathway of recovery between countries. However, we could not distinguish a difference in discharge destinations between the three MRC-SS groups. Alternatively, our population might have been more responsive to recovery. Herridge et al. [42] describe four disability subtypes based on the FIM seven days after ICU discharge as well as by age and length of ICU stay. These four subtypes determined a 1-year trajectory of recovery. We cannot directly compare the FIM due to the different time-points of the measurement (36 at ICU discharge in our cohort versus 54 one week after ICU discharge in [42]). However, when comparing baseline characteristics, we suspect that the shorter length of ICU stay in our cohort (5.9 versus 19.0 days in [42]), despite being slightly older (68 years versus 58 in [42]), might have been beneficial for recovery. Our results might also have been confounded by non-responders who might have been too ill to return the questionnaire or influenced by false negative results especially in the 'role physical' domain where severely weak patients (50 [IQR 25–75]) had surprisingly higher values than non-weak patients (25 [0–50]). Lastly, our sample might have been too small and selective to reveal a relevant association of ICUAW with quality of life.

Although the investigated risk factors were not inclusive of all previously known contributors to ICUAW [35], they support the findings of Fan et al. [12] who found that with every day

**Table 3. Univariate (crude) and multivariate (adjusted) regression models for chosen ICU risk factors with MRC-SS as response.**

| | n | Crude effect (95% CI) | p-value | n | Adjusted effect (95% CI) | p-value |
|---|---|---|---|---|---|---|
| **Gender** | | | | | | |
| *female* | | *Reference* | | | *Reference* | |
| **male** | 83 | 6.94 (1.61 to 12.27) | 0.013 | 80 | 5.51 (0.64 to 10.38) | 0.030 |
| **Group allocation** | | | | | | |
| *control group* | | *Reference* | | | *Reference* | |
| **experimental group** | 83 | -2.02 (-7.37 to 3.33) | 0.461 | 80 | -0.66 (-5.60 to 4.28) | 0.794 |
| **Illness severity** | | | | | | |
| **SOFA score** | 83 | -0.63 (-1.37 to 0.10) | 0.094 | 80 | -0.55 (-1.27 to 0.17) | 0.141 |
| **Length of ICU stay** | | | | | | |
| **Length of ICU stay at original hospital (days)** | 83 | -0.21 (-0.48 to 0.06) | 0.126 | 80 | -0.26 (-0.53 to 0.01) | 0.065 |
| **Activities of daily living (ADL)** | | | | | | |
| *Not restricted in ADL* | | *Reference* | | | *Reference* | |
| **Restricted in ADL** | 80 | -5.38 (-14.33 to 3.58) | 0.243 | 80 | -6.18 (-13.97 to 1.61) | 0.124 |
| **Mobilisation level in ICU** | | | | | | |
| *Mobilisation level in ICU: out-of-bed [a]* | | *Reference* | | | *Reference* | |
| **Mobilisation level in ICU: in-bed** | 83 | -23.80 (-37.05 to -10.54) | 0.001 | 80 | -24.57 (-37.03 to -12.11) | <0.001 |
| **Mobilisation level in ICU: edge-of-bed** | 83 | -6.89 (-12.22 to -1.55) | 0.013 | 80 | -7.20 (-12.78 to -1.62) | 0.014 |

[a] Overall p values for the factor "mobilisation level in the ICU" with Omnibus test: p = 0.001 for crude and p<0.001 for adjusted effect

Significance of the chosen variables in two regression models with MRC-SS as response. Crude regression is just response (MRC-SS) and corresponding explanatory variable. Adjusted regression includes all listed explanatory variables. The robust estimations give the same results with non-significant bias' tests (with the null: presence of the bias due to lack of robustness or outlying observations). None of the terms revealed non-linearity when fitting a regression model allowing non-linear dependence (via penalized splines) on explanatory variables. For full regression-output see supporting information (S1 File).

**Abbreviations:** SOFA = Sequential Organ Failure Assessment, ADL = activities of daily living

of bed rest muscle strength is reduced by 3–11% in critically ill ARDS patients. In-bed immobilisation is a largely modifiable risk factor [43] and might be targeted with therapeutic interventions in critically ill patients at risk. However, current evidence on the benefit of early mobilisation is conflicting and needs further study [44]. Our randomised controlled trial likewise failed to reveal a superior intervention to increase strength at ICU discharge when comparing two different exercise regimes [22]. However, early mobilisation was part of both trial arms and thus not under direct investigation. Overall, more research is needed on how to overcome this exposure outcome overlap and what kind of intervention should be prescribed. The influence of female gender on muscle strength has also previously been described [20] and might be a result of reduced muscle mass in women [35]. As a consequence, women might need different treatment from men. This could include early screening to identify weakness and might mean a prolonged or more intensive period of rehabilitation.

The strength of this study lies in the performance-based measurements—performed by trained physiotherapists—in a general ICU population with a high ICUAW incidence and adds relevant information about the functional short-term outcomes in generally, critically ill patients. Furthermore, the 6-month timeframe has a high clinical relevance because health-related quality of life appears to plateau after this period [45]. There are also several limitations. First, this secondary analysis was limited by the availability of the existing data and we did not adjust the significance threshold for multiple comparisons. Accordingly, additional risk factors may have remained undiscovered or the identified factors may have been a random association. Second, a rather high drop-out rate at six months (41%) substantially limits the possible conclusions about these patients' subsequent quality of life. Third, generalisability is limited

due to a selective and single-centre sample. Fourth, while it seemed that patients with ICUAW recovered, this observational study can draw no conclusions about possible interventions. Given that post-ICU rehabilitation presently lacks evidence to improve quality of life [46], the pathways of recovery need further investigation. A prospective mixed-methods cohort study might reveal possible interventions to target for future studies. Fifth, we may have underestimated the incidence of ICUAW because participants that could not be tested due to their inability to follow commands may have scored as weak or functionally impaired. The recommended cut-offs for a clinical ICUAW diagnosis as applied here are further limited by the absence of proper validation and the lack of a gold standard. For these reasons, to fully understand the consequences of ICUAW, more research on the diagnosis of ICUAW is needed. Finally, the retrospective and exploratory nature of our analysis requires further investigation. To the best of our knowledge, there are no publicly available datasets to validate our results on ICUAW risk factors or the functional outcome of patients after an ICUAW diagnosis. Our results should therefore be validated in a prospective, longitudinal observational study with long-term follow-up.

## Conclusions

Participants without ICUAW had superior functional performance at hospital discharge and shorter length of hospital stays when compared to participants with ICUAW. The increased strength was associated with early out-of-bed mobilisations during the ICU. However, after six months, participants with ICUAW reached similar health-related quality of life to participants without ICUAW at ICU discharge. This implies that recovery for critically ill, mechanically ventilated patients with ICUAW might be at least partly achieved. Nevertheless, these findings need to be validated in a prospective cohort study.

## Supporting information

**S1 Table. Timetable for primary and secondary outcome measures.**
(PDF)

**S2 Table. Baseline differences based upon completed versus missing MRC-SS.**
(PDF)

**S3 Table. Baseline differences based upon reason for missingness.**
(PDF)

**S4 Table. Sensitivity analysis comparing ICUAW versus non-ICUAW.** This includes German norm-based (1994) standardized sum-scores (T-values) for SF-36.
(PDF)

**S5 Table. Correlation-matrix for functional outcomes at hospital discharge and SF-36.**
(XLSX)

**S6 Table. Sensitivity analysis for the regression model either with or without the randomisation group or with or without the ADL variable.**
(PDF)

**S1 File. All regressions' output.**
(PDF)

## Acknowledgments

We sincerely thank all physiotherapists and ICU staff for their dedicated care; this study would not have been possible without them. We also thank Alan Haynes for the statistical advice and for reviewing the manuscript.

## Author Contributions

**Conceptualization:** Sabrina Eggmann, Gere Luder, Martin L. Verra, Irina Irincheeva, Caroline H. G. Bastiaenen, Stephan M. Jakob.

**Data curation:** Sabrina Eggmann.

**Formal analysis:** Sabrina Eggmann, Gere Luder, Irina Irincheeva.

**Funding acquisition:** Sabrina Eggmann.

**Investigation:** Sabrina Eggmann, Gere Luder, Martin L. Verra, Stephan M. Jakob.

**Methodology:** Sabrina Eggmann, Gere Luder, Martin L. Verra, Irina Irincheeva, Caroline H. G. Bastiaenen, Stephan M. Jakob.

**Project administration:** Sabrina Eggmann.

**Resources:** Sabrina Eggmann, Gere Luder, Martin L. Verra, Stephan M. Jakob.

**Software:** Sabrina Eggmann, Irina Irincheeva.

**Supervision:** Sabrina Eggmann, Martin L. Verra, Caroline H. G. Bastiaenen, Stephan M. Jakob.

**Validation:** Sabrina Eggmann.

**Visualization:** Sabrina Eggmann, Irina Irincheeva.

**Writing – original draft:** Sabrina Eggmann.

**Writing – review & editing:** Sabrina Eggmann, Gere Luder, Martin L. Verra, Irina Irincheeva, Caroline H. G. Bastiaenen, Stephan M. Jakob.

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
