## [Decision Letter · Decision Letter 0]

23 Dec 2019

PONE-D-19-28826

Functional ability and quality of life in critical illness survivors with intensive care unit acquired weakness: a secondary analysis of a randomised controlled trial

PLOS ONE

Dear Dr. Eggmann,

Thank you for submitting your manuscript to PLOS ONE. After careful consideration, we feel that it has merit but does not fully meet PLOS ONE’s publication criteria as it currently stands. Therefore, we invite you to submit a revised version  and much extended version of the manuscript that addresses the points raised during the review process.

This is a re-analysis of a published RCT which turns it into an small observation study. Thus, as also pointed out be the reviewers below the authors have to provide evidence, that their findings are robust with regard to e.g. multipicitiy issues that arise when one data set is used for exploration. As a consequence, at least two compulsory requirements arise: (1) the main findings have to be replicated in an independent cohort of similar design and (2) the authors should make core (i.e. the data reported in this paper jointly with the analysis code to reproduce the findings) anonymized patient-level of the study and the related staistical programming code available. If both requirements cannot be met, I do not see how the PLOS ONE quality criteria can be met.

We would appreciate receiving your revised manuscript by 02/01/2020. To enhance the reproducibility of your results, we recommend that if applicable you deposit your laboratory protocols in protocols.io, where a protocol can be assigned its own identifier (DOI) such that it can be cited independently in the future. For instructions see: http://journals.plos.org/plosone/s/submission-guidelines#loc-laboratory-protocols

We look forward to receiving your revised manuscript.

Kind regards,

André Scherag, Ph.D.

Academic Editor

PLOS ONE

Journal Requirements:

Reviewers' comments:

Reviewer's Responses to Questions

**Comments to the Author**

1. Is the manuscript technically sound, and do the data support the conclusions?

Reviewer #1: Yes

Reviewer #2: Partly

2. Has the statistical analysis been performed appropriately and rigorously? 

Reviewer #1: Yes

Reviewer #2: Yes

3. Have the authors made all data underlying the findings in their manuscript fully available?

Reviewer #1: Yes

Reviewer #2: Yes

4. Is the manuscript presented in an intelligible fashion and written in standard English?

Reviewer #1: Yes

Reviewer #2: Yes

5. Review Comments to the Author

Reviewer #1: A secondary analysis of data from a prior randomized trial in which exercise related outcomes did not differ was used to compare intensive care unit acquired weakness (ICUAW) categorized as severe,  moderate or none. The primary outcomes were the 6-Minute Walk Test and Functional Independence Measure at hospital discharge. Secondary outcomes included health-related quality of life after six months. At discharge outcomes differed significantly, but at 6 months the outcomes were not statistically different.

Minor revisions:

1-     Lines 223 & 225: For clarity, replace 25% and 75% with first and third quartiles or simply indicate interquartile range.

2-    Table 3: (A) For better clarity in presentation, consider adding a header row for each factor. For instance, above female/male add a row with the word “gender” in the first column. (B) For factors with more than two levels include the overall p-value for each factor rather than pairwise p-values. (C) In the title, clarify that crude results imply from univariate models and adjusted results are from a multivariate model.

Reviewer #2: ICUAW might put ICU-survivors on increased risk for long-term functional impairment and increased mortality. Too little is known about risk-factors, correlates and long-term consequences of ICUAW. The authors present a prospective cohort study with a six-month follow-up, that is conducted as a secondary analysis of the data of a RCT. The study is limited by a rather small study sample and a follow-up period of only six months. While the study questions are of high importance, I see some serious flaws in the conducted analyses that require revise.

My major concern is, what measures were used as predictors, and what measures where used as primary outcomes. While the primary outcomes of the RCT were measured at hospital discharge, the primary focus of this cohort study should be on the long-term outcomes measured at six months, since these are of higher relevance for patients. Specific subscales of the SF-36 that are most likely affected by ICUAW should be regarded as outcomes of primary interest (e.g. physical functioning, physical role functioning).

Concerning MRC-test and 6-min walk test, it is hard to distinguish these two as measures of two different “illness entities”, since Jolley et al (2016, Chest) discuss the 6-Min. walk test as an indicator of ICUAW in the early recovery period. The measurement at hospital discharge might also be the better choice to identify patients that have ICUAW, since the measurement at ICU discharge might still reflect aspects of the acute critical illness that might be highly unstable till the early recovery phase is achieved.

In my understanding, there are several indicators of ICUAW in the data of the RCT (MRC test, 6 min. walk test, grip strength etc.), some measured at ICU, other at hospital discharge. All of these might be suitable to predict 6-month follow-up status in an exploratory approach. The interrelatedness of all these indicators of ICUAW should be presented (maybe as a correlation-matrix in the supplement).

Although the FIM cannot be considered a direct indicator of ICUAW it would be of interest to see, if it is able to predict respective functional subscales of the SF36 at 6 months.

Consider to also predict the 6-min walk test (as an indicator of ICUAW during early recovery) from baseline and ICU-related risk-factors.

General comments:

I would recommend not to use categorized predictor variables (e.g. three categories of MRC-SS) in such a small study sample, since this severely can reduce power. Use continuous variables when possible and use categorization only for descriptive analysis.

If hospital length of stay should be considered as an outcome, then only the days after ICU-discharge should be counted.

T-values for SF-36 are calculated based on US-norms from 1990. There must be more recent and appropriate Norms available (e.g. there do exist recent German norm values).

Given the small sample size, multivariable analysis should not use to many variables. I suggest using a method that reduces the number of variables for multivariate analysis (e.g. only significant in univariate analysis).

To help interpret the results of the study, some considerations should be added on how big the test power was to detect relations between ICUAW indicators and SF-36 scales at 6-month follow up (referring to reasonable effect sizes obtained from earlier studies). It might be, that the power of the study was too small to find effects comparable to those that were previously reported. Based on such considerations “insignificant” findings in comparison to other studies should not be over-interpreted in the discussion. All results should be interpreted with caution because of the explorative nature of the study.

Authors stated that all data are fully available but did not report where.

6. PLOS authors have the option to publish the peer review history of their article (what does this mean?). If published, this will include your full peer review and any attached files.

Reviewer #1: No

Reviewer #2: No

---

## [Author Response · Author response to Decision Letter 0]

1 Feb 2020

Responses to Reviewers

The reviewers’ comments are shown in bold type, our responses in plain type and changed passages of the manuscript are shown in cursive type. All pages and lines are in reference to the unmarked manuscript version: titled “Manuscript”.

Reviewer #1

A secondary analysis of data from a prior randomized trial in which exercise related outcomes did not differ was used to compare intensive care unit acquired weakness (ICUAW) categorized as severe, moderate or none. The primary outcomes were the 6-Minute Walk Test and Functional Independence Measure at hospital discharge. Secondary outcomes included health-related quality of life after six months. At discharge outcomes differed significantly, but at 6 months the outcomes were not statistically different.

We thank the reviewer for taking the time to review our manuscript and are grateful for the constructive and valuable feedback. All comments have been answered below.

Minor revisions:

1- Lines 223 & 225: For clarity, replace 25% and 75% with first and third quartiles or simply indicate interquartile range.

Tables 1 and 2 have been adapted as suggested by the reviewer to indicate interquartile range from 25% to 75%. We further clarified the section in the statistical analysis:

Page 6, lines 165-167:

“Continuous data could not be assumed normally distributed and are therefore presented as medians with interquartile ranges (IQR: first (25%) quartile to third (75%) quartile).”

2- Table 3: (A) For better clarity in presentation, consider adding a header row for each factor. For instance, above female/male add a row with the word “gender” in the first column. (B) For factors with more than two levels include the overall p-value for each factor rather than pairwise p-values. (C) In the title, clarify that crude results imply from univariate models and adjusted results are from a multivariate model.

(A) A header row has been added for each factor in Table 3 (page 13)

(B) We added a footnote for overall p-values, analysed with the Omnibus test, for the factor “mobilisation level in the ICU” (page 14, lines 295-296):

 “a Overall p values for the factor “mobilisation level in the ICU” with Omnibus test: p=0.001 for crude and p<0.001 for adjusted effect”

(C) The title of Table 3 has been changed to:

Page 13, lines 292-293:

“Univariate (crude) and multivariate (adjusted) regression models for chosen ICU risk factors with MRC-SS as response.”

Reviewer #2

ICUAW might put ICU-survivors on increased risk for long-term functional impairment and increased mortality. Too little is known about risk-factors, correlates and long-term consequences of ICUAW. The authors present a prospective cohort study with a six-month follow-up, that is conducted as a secondary analysis of the data of a RCT. The study is limited by a rather small study sample and a follow-up period of only six months. While the study questions are of high importance, I see some serious flaws in the conducted analyses that require revise.

We thank the reviewer for taking the time to review our manuscript and are grateful for the constructive and valuable feedback. All comments have been answered below.

My major concern is, what measures were used as predictors, and what measures where used as primary outcomes. While the primary outcomes of the RCT were measured at hospital discharge, the primary focus of this cohort study should be on the long-term outcomes measured at six months, since these are of higher relevance for patients. Specific subscales of the SF-36 that are most likely affected by ICUAW should be regarded as outcomes of primary interest (e.g. physical functioning, physical role functioning).

We acknowledge the reviewer’s concern that we did not focus sufficiently on long-term outcomes (SF-36). All subscales of the SF-36 are reported in Table 2, but to increase their visibility, we added a graph (Fig. 4) for the physical and mental health sum-scores of the SF-36. The manuscript has been changed (page 10, lines 243-245):

“Fig 3 further illustrates the distribution between the three MRC-SS groups for the Timed ‘Up & Go’ test and hospital length of stay after ICU discharge, while Fig 4 illustrates the distribution for the SF-36 physical and mental health sum-scores.”

Fig 4 title and legend (page 12, lines 280-282):

“Fig 4. Illustration of the SF-36 physical and mental health sum-scores per MRC-SS group.

Illustration of the SF-36 physical health sum-score (a) and mental health sum-score (b) with non-parametric Cuzick test accepting the null hypothesis of equal distributions.”

We further provide a correlation-matrix on the functional outcomes measured at hospital discharge with the SF-36 in the supplement (S5 Table) as suggested by the reviewer in one of the following questions.

We agree, that in principal the focus should be more on long-term outcomes. However, we a-priori chose to use the same short-term outcomes as in the original trial, because these were the endpoints with the least drop-outs and thus the most likely to be sufficiently powered. We therefore would not feel confident to change primary outcomes at this point. We added this limitation to the manuscript:

Page 16, lines 357-358:

“Lastly, our sample might have been too small and selective to reveal a relevant association of ICUAW with quality of life.”

Page 17, lines 393-394:

“Our results should therefore be validated in a prospective, longitudinal observational study with long-term follow-up.”

Concerning MRC-test and 6-min walk test, it is hard to distinguish these two as measures of two different “illness entities”, since Jolley et al (2016, Chest) discuss the 6-Min. walk test as an indicator of ICUAW in the early recovery period. The measurement at hospital discharge might also be the better choice to identify patients that have ICUAW, since the measurement at ICU discharge might still reflect aspects of the acute critical illness that might be highly unstable till the early recovery phase is achieved.

We agree that measuring weakness at ICU discharge is likely to be unstable and still influenced by the critical illness, which is exactly why we were interested in the subsequent outcome. As Jolley et al. (2016) discussed, there is very few data to support the assumption that ICUAW impacts subsequent physical function [1]. We therefore set out to explore this. 

We also agree that the measurements at hospital discharge might provide additional important diagnostic/predictive value. However, we think it is important to distinguish that the 6MWT and MRC-SS test measure two rather different constructs as described by the WHO framework “International Classification of Functioning, Disability and Health” (ICF) for measuring health and disability [2]. Accordingly, the 6MWT is allocated to the domains of body functions (exercise tolerance function b455) and activities and participation (mobility d4 and walking short distances d4500) [3]. In contrast the MRC-SS is allocated to the domains of body functions (muscle functions b730-b749). Our data somewhat confirms this hypothesis through a low, positive correlation between the 6MWT and the MRC-SS (Spearman rho r=0.316, p=0.006). We think the second construct (MRC-SS) a better fit to identify ICUAW given its official diagnosis as a “generalised weakness developing after the onset of critical illness” [4: page 303]. We therefore chose to use the same diagnostic approach as suggested by others [4-6]. We acknowledge that the MRC-SS has its limitations due to its reliance on patient cooperation as stated in our manuscript (page 17, lines 387-390), but we believe that identification of patients with ICUAW should occur as early as possible – not only at hospital discharge. Indeed, our results seem to support an early diagnosis given its association with worsened short-term functional outcome.

Nevertheless, the 6MWT could have some diagnostic/predictive value for long-term outcomes. We do not know of any such previous studies. However, our data is probably underpowered to sufficiently investigate this. A correlation of the 6MWT and the FIM with the physical health sum-score of the SF-36 did not reveal any relevant correlation (6MWT: r=0.199, p=0.153, n=53, FIM: r=-0.088, p=0.533, n=53). We therefore did not explore this any further.

In my understanding, there are several indicators of ICUAW in the data of the RCT (MRC test, 6 min. walk test, grip strength etc.), some measured at ICU, other at hospital discharge. All of these might be suitable to predict 6-month follow-up status in an exploratory approach. The interrelatedness of all these indicators of ICUAW should be presented (maybe as a correlation-matrix in the supplement).

We agree with the reviewer, that it would be interesting to explore 6-month follow-up status based on functional outcomes (6MWT, FIM, Timed ‘Up & Go’ test, grip strength, quadriceps strength and MRC-SS) measured during the hospital stay. The manuscript has been revised and as suggested we added a correlation-matrix as a supplement (new S5 Table, the previous S5 Table has been renamed S6 Table):

Page 6, lines 158-160:

“We further explored whether functional performance at hospital discharge might be useful to predict 6-month quality of life.”

Page 7, lines 182-183:

“Correlations were investigated with non-parametric Spearman correlation coefficients.”

Page 10, lines 248-249:

“Functional performance at hospital discharge was not associated with quality of life (S5 Table).”

Page 14, lines 311-313:

However, contrary to our hypothesis, quality of life after six months was similar in participants with severe, moderate or no ICUAW and not associated with functional performance in the hospital.

The results seem likely to be underpowered due to the high drop-out rates at 6-month follow-up (28% 6-month mortality). This limits our conclusions substantially and has been discussed as a limitation: 

Page 16, lines 357-358:

“Lastly, our sample might have been too small and selective to reveal a relevant association of ICUAW with quality of life.”

Although the FIM cannot be considered a direct indicator of ICUAW it would be of interest to see, if it is able to predict respective functional subscales of the SF36 at 6 months.

Indeed, there are studies that found that the FIM may be an excellent indicator of recovery [7]. The findings of this study can unfortunately not be directly compared to our results, but has been discussed on page 15, lines 347-353.

Given the newly added exploratory correlation-matrix in S5 Table, our data do not indicate that the FIM is able to predict any of the functional subscales of the SF-36 (correlations non-significant with r<0.2). Again, this might be a result of the high drop-out rates at six months (28% 6-month mortality in the whole cohort), which require us to be careful in interpreting these data. To make the limitations of an insufficiently powered 6-month outcome absolutely clear to the reader, we aggravated this major limitation in the manuscript:

Page 16, lines 357-358:

“Lastly, our sample might have been too small and selective to reveal a relevant association of ICUAW with quality of life.”

Page 16, lines 379-381:

“Second, a rather high drop-out rate at six months (41%) substantially limits the possible conclusions about these patients’ subsequent quality of life.”

Consider to also predict the 6-min walk test (as an indicator of ICUAW during early recovery) from baseline and ICU-related risk-factors.

This is an interesting point. As stated previously we believe that ICUAW should be identified as early as possible, the same would of course apply to functional disability (e.g. 6MWT at hospital discharge). We therefore explored the influence of the same baseline and ICU-related risk-factors on their influence on the 6MWT with a univariate (crude) and multivariate (adjusted) regression model. 

We report the results here. None of the investigated risk factors were associated with 6MWT. This is somewhat surprising because normative values for females are generally lower than for males [8]. Nevertheless, we did not report results in the manuscript because we feel that this analysis is too exploratory, has not been previously planned and did probably not consider all relevant factors from a theoretical perspective. Recent evidence for example indicates that pre-existing comorbidities and ARDS patients (versus non-ARDS) might be much better predictors [9].

Univariate (crude) and multivariate (adjusted) regression models for chosen ICU risk factors with 6MWT as response,

 n Crude effect (95% CI) p-value n Adjusted effect (95% CI) p-value

Gender

female Reference Reference 

male 84 56.31 (-12.00 to 124.63) 0.110 82 58.74 (-11.34 to 128.82) 0.105

Group allocation

control group Reference Reference 

experimental group 84 -18.58 (-85.94 to 48.78) 0.590 82 -1.93 (-73.78 to 69.91) 0.958

Illness severity

SOFA score 84 -8.34 (-17.53 to 0.84) 0.079 82 -7.81 (-17.83 to 2.20) 0.130

Length of ICU stay

Length of ICU stay at original hospital (days) 84 -1.84 (-5.31 to 1.63) 0.301 82 -1.57 (-5.43 to 2.29) 0.427

Activities of daily living (ADL)

Not restricted in ADL Reference Reference 

Restricted in ADL 82 -49.47 (-171.69 to 72.74) 0.430 82 -60.89 (-184.43 to 62.64) 0.337

Mobilisation level in ICU

Mobilisation level in ICU: out-of-bed Reference Reference 

Mobilisation level in ICU: in-bed 84 -114.42 (-237.45 to 8.61) 0.072 82 -111.50 (-235.58 to 12.59) 0.082

Mobilisation level in ICU: edge-of-bed 84 -33.40 (-104.92 to 38.12) 0.363 82 -21.81 (-103.70 to 60.08) 0.603

Table legend: Significance of the chosen variables in two regression models with 6MWT as response. Crude regression is just response (6MWT) and corresponding explanatory variable. Adjusted regression includes all listed explanatory variables. The robust estimations give the same results with non-significant bias’ tests (with the null: presence of the bias due to lack of robustness or outlying observations). None of the terms revealed non-linearity when fitting a regression model allowing non-linear dependence (via penalized splines) on explanatory variables.

General comments:

I would recommend not to use categorized predictor variables (e.g. three categories of MRC-SS) in such a small study sample, since this severely can reduce power. Use continuous variables when possible and use categorization only for descriptive analysis.

We agree with the reviewer that three categories of the MRC-SS would reduce power if analysed with Kruskal-Wallis test. However, the Cuzick test can test an ordered relationship (e.g. severe is worse than moderate which is worse than no weakness plus severe is worse than no weakness) without multiple tests [10]. It is therefore more powerful than the Kruskal-Wallis test, and this test has been recommended by our statistician (I.I.). It should be acknowledged, that MRC-SS is not a truly linear construct, since the differences between any two adjacent numbers are not necessarily equal. Furthermore, the cut-off for ICUAW has not been properly validated as described in our manuscript on page 17, lines 387-389. We conducted a sensitivity analysis comparing all participants with ICUAW versus non-ICUAW (see page 7, lines 185-188). Results are presented in S4 Table and described on page 10, lines 245-248:

“Sensitivity analysis comparing all participants with ICUAW versus non-ICUAW similarly confirmed a significant association of ICUAW with functional disability and prolonged hospital stay, and lack of association with 6-month health-related quality of life (S4 Table).”

If hospital length of stay should be considered as an outcome, then only the days after ICU-discharge should be counted.

The reviewer raises an important point. We corrected this outcome to include only days after ICU discharge. Accordingly, Table 2, Fig 3 and S4 table have been changed to report the new results, which do not change our results/conclusions.

Variable n All with complete MRC-SS n Severe weakness n Moderate weakness n No weakness p-value

Hospital length of stay after ICU discharge (days) 83 16.87 [11.16 - 26.92] 17 20.9 [15.83 - 30.73] 32 16.86 [13.07 - 27.10] 34 11.16 [7.35 - 19.74] 0.008

We further made the following changes to the manuscript:

Abstract, page 2, lines 44-48:

“Functional outcomes and length of hospital stay significantly differed in patients with severe, moderate to no weakness (6-Minute Walk test: p=0.013; 110m [IQR 75-240], 196m [90-324.25], 222.5m [129-378.75], Functional Independence Measure: p=0.001; 91[IQR 68-101], 113[102.5-118.5], 112[97-123], length of stay after ICU discharge: p=0.008; 20.9d [IQR 15.83-30.73], 16.86d [13.07-27.10], 11.16d [7.35-19.74]).”

Page 10, lines 234-236:

“Weakness at ICU discharge was significantly associated with functional disability at hospital discharge and subsequent length of hospital stay, but not with health-related quality of life after six months.”

T-values for SF-36 are calculated based on US-norms from 1990. There must be more recent and appropriate Norms available (e.g. there do exist recent German norm values).

Indeed, there are German norm values from 1994 available. However, the official manual of the SF-36 version 2 [11] explicitly recommends to use US normative values for studies not from Germany or to compare study results internationally: “CAUTION: To COMPARE your own results of the SF-36 with the results in present international or German publications, you have to calculate the summary scales for the SF-36 based on the US POPULATION NORM 1990!” [cited directly from the SPSS syntax]. Moreover, US normative values are comparable to German normative values [12]. We therefore chose to report US norm values in the original trial as well as for this manuscript. Nevertheless, we re-calculated German norm-based standardized sum-scores (T-values).

Comparisons per MRC-SS group with the Cuzick-test did not change results:

Variable n All with complete MRC-SS n Severe weakness n Moderate weakness n No weakness p-value

Physical health (sum-score) based on German-norm 1994 49 42.67 [36.54 - 49.56] 12 43.79 [33.96 - 47.72] 13 42.77 [29.02 - 48.91] 24 42.32 [37.57 - 49.59] 0.922

Mental health (sum-score) based on German-norm 1994 49 47.59 [41.69 - 55.54] 12 47.49 [42.79 - 58.09] 13 43.3 [32.87 - 48.88] 24 48.55 [43.08 - 55.59] 0.884

Given the above warnings, we report German normative values only in the sensitivity analysis in the supporting information (see S4 Table).

The following changes were made to the manuscript:

Page 12, lines 264-265 (Table 2 legend)

“SF-36 (version 2): worst score: 0, best score: 100, sum-score: T-values where the population mean is 50 and the SD is 10; based on US-population 1990. German norm-based (1994) standardized sum-scores (T-values) for SF-36 were similar to the US-population (data shown in sensitivity analysis in S4 Table).”

Page 23, lines 594-595 (description of the supporting information):

“S4 Table. Sensitivity analysis comparing ICUAW versus non-ICUAW. This includes German norm-based (1994) standardized sum-scores (T-values) for SF-36.”

Given the small sample size, multivariable analysis should not use to many variables. I suggest using a method that reduces the number of variables for multivariate analysis (e.g. only significant in univariate analysis).

We acknowledge the reviewers concern and agree that we should be cautious of overfitting. However, models were defined a priori on theoretical grounds based on previous evidence. Since point estimates are largely similar and confidence intervals overlap to a large degree, we feel that our adjusted models are reasonable, although they should be interpreted with caution. Additionally, there is a rule-of-thumb in linear regression that there should be at least 10 observations per model parameter [13]. With at least 80 observations and no more than 6 model parameters, this rule-of-thumb is satisfied.

To help interpret the results of the study, some considerations should be added on how big the test power was to detect relations between ICUAW indicators and SF-36 scales at 6-month follow up (referring to reasonable effect sizes obtained from earlier studies). It might be, that the power of the study was too small to find effects comparable to those that were previously reported. Based on such considerations “insignificant” findings in comparison to other studies should not be over-interpreted in the discussion. All results should be interpreted with caution because of the explorative nature of the study.

We fully agree with the reviewer that the power to detect an effect of ICUAW on 6-month outcomes was probably too low and that all results should be interpreted with caution because of the explorative nature of the study. However, we strongly believe – and are reinforced by our statistician – that power analysis is for study planning and not useful as a tool to justify nonsignificant results because the power is directly related to the p-value as discussed by several authors [14-16].

We concede that our results might have been overinterpreted and therefore revised our limitations and stressed the need to confirm these results in a prospective clinical trial. We further searched for a data set that might help to validate our results. Unfortunately, we were unable to find such a publicly available data set. We therefore re-formulated the following limitations and conclusions of our manuscript:

Abstract, page 3, lines 57:

“Prospective trials are needed to validate these results.”

Page 14, lines 316-318:

“These results provide useful information about the functional ability and subsequent health-related quality of life in a general, mechanically ventilated ICU population after a critical illness period, but need to be validated in prospective studies.”

Page 14, lines 335-336:

“When comparing our results for health-related quality of life with the existing literature, we found some discrepancies.”

Page 15-16, lines 354-258:

“Our results might also have been confounded by non-responders who might have been too ill to return the questionnaire or influenced by false negative results especially in the ‘role physical’ domain where severely weak patients (50 [IQR 25-75]) had surprisingly higher values than non-weak patients (25 [0-50]). Lastly, our sample might have been too small and selective to reveal a relevant association of ICUAW with quality of life.”

Page 16, lines 378-381:

“Accordingly, additional risk factors may have remained undiscovered or the identified factors may have been a random association. Second, a rather high drop-out rate at six months (41%) substantially limits the possible conclusions about these patients’ subsequent quality of life.”

Page 17, lines 390-394:

“Finally, the retrospective and exploratory nature of our analysis requires further investigation. To the best of our knowledge, there are no publicly available datasets to validate our results on ICUAW risk factors or the functional outcome of patients after an ICUAW diagnosis. Our results should therefore be validated in a prospective, longitudinal observational study with long-term follow-up.”

Page 17, lines 403-404:

“Nevertheless, these findings need to be validated in a prospective cohort study.”

Authors stated that all data are fully available but did not report where.

We apologise for this miscommunication. Our data availability statement has been entered separately into the Editorial Manager. All data including codes will be accessible from the “Bern Open Repository and Information System”. If the manuscript is accepted, data and codes will be uploaded. This ensures that any amendments made during the review process are included in the codes and data. The assigned DOI will subsequently be forwarded to the journal. The statement is as follows:

“The dataset and the analysis script are available from the “Bern Open Repository and Information System” database (https://boris.unibe.ch/) under the DOI XXXXX.”

References:

1. Adhikari NK, Fowler RA, Bhagwanjee S, Rubenfeld GD. Critical care and the global burden of critical illness in adults. Lancet. 2010;376(9749):1339-46. Epub 2010/10/12. doi: 10.1016/s0140-6736(10)60446-1. PubMed PMID: 20934212.

2. Lee CM, Fan E. ICU-acquired weakness: what is preventing its rehabilitation in critically ill patients? BMC medicine. 2012;10:115. Epub 2012/10/05. doi: 10.1186/1741-7015-10-115. PubMed PMID: 23033976; PubMed Central PMCID: PMC3520774.

3. Herridge MS, Tansey CM, Matte A, Tomlinson G, Diaz-Granados N, Cooper A, et al. Functional disability 5 years after acute respiratory distress syndrome. The New England journal of medicine. 2011;364(14):1293-304. Epub 2011/04/08. doi: 10.1056/NEJMoa1011802. PubMed PMID: 21470008.

4. Stevens RD, Marshall SA, Cornblath DR, Hoke A, Needham DM, de Jonghe B, et al. A framework for diagnosing and classifying intensive care unit-acquired weakness. Critical care medicine. 2009;37(10 Suppl):S299-308. Epub 2010/02/06. doi: 10.1097/CCM.0b013e3181b6ef67. PubMed PMID: 20046114.

5. Parker AM, Sricharoenchai T, Raparla S, Schneck KW, Bienvenu OJ, Needham DM. Posttraumatic stress disorder in critical illness survivors: a metaanalysis. Critical care medicine. 2015;43(5):1121-9. Epub 2015/02/06. doi: 10.1097/ccm.0000000000000882. PubMed PMID: 25654178.

6. Cuthbertson BH, Elders A, Hall S, Taylor J, MacLennan G, Mackirdy F, et al. Mortality and quality of life in the five years after severe sepsis. Critical care (London, England). 2013;17(2):R70. Epub 2013/04/17. doi: 10.1186/cc12616. PubMed PMID: 23587132; PubMed Central PMCID: PMCPMC4057306.

7. Weijs PJ, Looijaard WG, Dekker IM, Stapel SN, Girbes AR, Oudemans-van Straaten HM, et al. Low skeletal muscle area is a risk factor for mortality in mechanically ventilated critically ill patients. Critical care (London, England). 2014;18(2):R12. Epub 2014/01/15. doi: 10.1186/cc13189. PubMed PMID: 24410863; PubMed Central PMCID: PMCPmc4028783.

8. Enright PL, Sherrill DL. Reference equations for the six-minute walk in healthy adults. American journal of respiratory and critical care medicine. 1998;158(5 Pt 1):1384-7. Epub 1998/11/17. doi: 10.1164/ajrccm.158.5.9710086. PubMed PMID: 9817683.

9. Parry SM, Nalamalapu SR, Nunna K, Rabiee A, Friedman LA, Colantuoni E, et al. Six-Minute Walk Distance After Critical Illness: A Systematic Review and Meta-Analysis. Journal of intensive care medicine. 2019:885066619885838-. doi: 10.1177/0885066619885838. PubMed PMID: 31690160.

10. Dinglas VD, Aronson Friedman L, Colantuoni E, Mendez-Tellez PA, Shanholtz CB, Ciesla ND, et al. Muscle Weakness and 5-Year Survival in Acute Respiratory Distress Syndrome Survivors. Critical care medicine. 2017;45(3):446-53. Epub 2017/01/10. doi: 10.1097/ccm.0000000000002208. PubMed PMID: 28067712; PubMed Central PMCID: PMCPmc5315580.

11. Morfeld M, Kirchberger, I., Bullinger, M. SF-36: Fragebogen zum Gesundheitszustand.

12. Ellert U, Kurth B-M. Methodische Betrachtungen zu den Summenscores des SF-36 anhand der erwachsenen bundesdeutschen Bevölkerung. Bundesgesundheitsblatt - Gesundheitsforschung -Gesundheitsschutz. 2004;47(11):1027-32. doi: 10.1007/s00103-004-0933-1.

13. Harrell FE, Jr., Lee KL, Mark DB. Multivariable prognostic models: issues in developing models, evaluating assumptions and adequacy, and measuring and reducing errors. Statistics in medicine. 1996;15(4):361-87. Epub 1996/02/28. doi: 10.1002/(sici)1097-0258(19960229)15:4<361::aid-sim168>3.0.co;2-4. PubMed PMID: 8668867.

14. Hoenig JM, Heisey DM. The Abuse of Power. The American Statistician. 2001;55(1):19-24. doi: 10.1198/000313001300339897.

15. Lenth RV. The University of Iowa 2007 [01.02.2020]. Available from: https://stat.uiowa.edu/sites/stat.uiowa.edu/files/techrep/tr378.pdf.

16. Goodman SN, Berlin JA. The use of predicted confidence intervals when planning experiments and the misuse of power when interpreting results. Annals of internal medicine. 1994;121(3):200-6. Epub 1994/08/01. doi: 10.7326/0003-4819-121-3-199408010-00008. PubMed PMID: 8017747.

---

## [Decision Letter · Decision Letter 1]

13 Feb 2020

Functional ability and quality of life in critical illness survivors with intensive care unit acquired weakness: a secondary analysis of a randomised controlled trial

PONE-D-19-28826R1

Dear Dr. Eggmann,

We are pleased to inform you that your manuscript has been judged scientifically suitable for publication and will be formally accepted for publication once it complies with all outstanding technical requirements.

With kind regards,

André Scherag, Ph.D.

Academic Editor

PLOS ONE

Additional Editor Comments (optional):

Reviewers' comments:

Reviewer's Responses to Questions

**Comments to the Author**

1. If the authors have adequately addressed your comments raised in a previous round of review and you feel that this manuscript is now acceptable for publication, you may indicate that here to bypass the “Comments to the Author” section, enter your conflict of interest statement in the “Confidential to Editor” section, and submit your "Accept" recommendation.

Reviewer #1: All comments have been addressed

Reviewer #2: All comments have been addressed

2. Is the manuscript technically sound, and do the data support the conclusions?

Reviewer #1: (No Response)

Reviewer #2: Yes

3. Has the statistical analysis been performed appropriately and rigorously? 

Reviewer #1: (No Response)

Reviewer #2: Yes

4. Have the authors made all data underlying the findings in their manuscript fully available?

Reviewer #1: (No Response)

Reviewer #2: Yes

5. Is the manuscript presented in an intelligible fashion and written in standard English?

Reviewer #1: (No Response)

Reviewer #2: Yes

6. Review Comments to the Author

Reviewer #1: The authors thoroughly and clearly addressed my prior comments. Thank-you!

Reviewer #2: The authors have invested a lot of effort in improving the manuscript. Thank you very much for this.

My only recommendation left is a change to the title:

"Functional ability and quality of life in critical illness survivors with intensive care unit acquired weakness: a secondary analysis of a randomised controlled trial"

This reads as if only patients with weakness were included. A possible change could be:

"Effects of intensive care unit acquired weakness on functional ability and quality of life in critical illness survivors: a secondary analysis of a randomised controlled trial"

7. PLOS authors have the option to publish the peer review history of their article (what does this mean?). If published, this will include your full peer review and any attached files.

Reviewer #1: No

Reviewer #2: No

---

## [Editor Report · Acceptance letter]

20 Feb 2020

PONE-D-19-28826R1 

Functional ability and quality of life in critical illness survivors with intensive care unit acquired weakness: a secondary analysis of a randomised controlled trial 

Dear Dr. Eggmann:

I am pleased to inform you that your manuscript has been deemed suitable for publication in PLOS ONE. Congratulations! Your manuscript is now with our production department. 

With kind regards,

on behalf of

Prof. Andre Scherag 

Academic Editor

PLOS ONE